# Free and Esterified Tocopherols, Tocotrienols and Other Extractable and Non-Extractable Tocochromanol-Related Molecules: Compendium of Knowledge, Future Perspectives and Recommendations for Chromatographic Techniques, Tools, and Approaches Used for Tocochromanol Determination

**DOI:** 10.3390/molecules27196560

**Published:** 2022-10-04

**Authors:** Paweł Górnaś, Georgijs Baškirovs, Aleksander Siger

**Affiliations:** 1Institute of Horticulture, Graudu 1, LV-3701 Dobele, Latvia; 2Department of Food Biochemistry and Analysis, Poznan University of Life Sciences, Wojska Polskiego 48, 60-637 Poznan, Poland

**Keywords:** tocopherols, tocotrienols, tocochromanols, tocols, vitamin E, LC, SFC, GC

## Abstract

Free and esterified (bound) tocopherols, tocotrienols and other tocochromanol-related compounds, often referred to “tocols”, are lipophilic antioxidants of great importance for health. For instance, α-tocopherol is the only tocochromanol with vitamin E activity, while tocotrienols have a positive impact on health and are proposed in the prevention and therapy of so-called modern diseases. Tocopherols, tocotrienols and plastochromanol-8 are the most well-known tocochromanols; in turn, knowledge about tocodienols, tocomonoenols, and other rare tocochromanol-related compounds is limited due to several challenges in analytical chemistry and/or low concentration in plant material. The presence of free, esterified, and non-extractable tocochromanols in plant material as well as their biological function, which may be of great scientific, agricultural and medicinal importance, is also poorly studied. Due to the lack of modern protocols as well as equipment and tools, for instance, techniques suitable for the efficient and simultaneous chromatographical separation of major and minor tocochromanols, the topic requires attention and new solutions, and/or standardization, and proper terminology. This review discusses the advantages and disadvantages of different chromatographic techniques, tools and approaches used for the separation and detection of different tocochromanols in plant material and foodstuffs. Sources of tocochromanols and procedures for obtaining different tocochromanol analytical standards are also described. Finally, future challenges are discussed and perspective green techniques for tocochromanol determination are proposed along with best practice recommendations. The present manuscript aims to present key aspects and protocols related to tocochromanol determination, correct identification, and the interpretation of obtained results.

## 1. Introduction

Tocochromanol history starts in 1922 when Bishop and Evans recognized α-tocopherol (α-T) as a pregnancy-supporting molecule in rats [1]. In 1937 α-, β-, and γ-T were isolated from different plant sources and had different efficacy in preventing vitamin E avitaminosis in rats [2]. Since then, 100 years have passed, but it remains unclear whether any other tocochromanol molecule can prevent vitamin E avitaminosis in humans the way α-T does. According to Azzi [3], the term should not be used interchangeably for different tocochromanols/tocols, since only α-T meets the criteria of preventing vitamin E ataxia, which is the disease caused by a vitamin E deficiency in humans. Tocopherols (Ts), especially α and γ homologues, are common in nature [4]. Tocotrienols (T3s) and other tocochromanol-related compounds are less known and often are not investigated. Hence, fewer reports can be found on their identification, detection, taxonomic distribution, biosynthesis, metabolism and biological function, while those molecules can have significant health beneficial properties [5]. It is estimated that only about 3% of research and clinical trials on the impact of tocochromanol supplementation on health concerns T3s, while 97% is dedicated to Ts [6], although several reports indicate superior biological properties of T3s to Ts, such as antioxidant and anti-inflammatory activities as well as their preventive effect against cancer, diabetes, cardiovascular and neurodegenerative diseases [6,7,8]. In the case of α-T, there is a clear lack of cancer-preventive activity. As for δ-, γ-T, and T3s, the effect on cancer risk is at least unclear and requires more studies [9]. Another topic worth highlighting is ester-bound tocochromanols, which are non-fissile in the human gastrointestinal tract, and their potential biological activity is unknown [10]. Esterified tocochromanols have been found, e.g., in walnuts, cucumber, chili, and bell pepper [10]. Since the majority of tocochromanol content determination methods in plant material use a standard protocol of saponification, during which bound tocochromanols are liberated (converted to free) from bound forms, knowledge about bound tocochromanols and relations/ratio between free and bound forms in nature is limited. Saponification is the most used protocol for tocochromanol determination due to higher tocol recovery compared to other methods. Other extraction techniques are less efficient, which is possibly due to the presence of some tocochromanols in ester forms or as compounds bound to the matrix, which are difficult to extract by tested solvents and techniques [11]. The topic of extractable and non-extractable tocochromanols has not been investigated. The challenge with the determination of esterified tocochromanols is not only associated with the procedure of tocol determination but also with lack of standards. Similarly, commercial standards for some other tocochromanols are not available, e.g., the well-known plastochromanol-8 (PC-8) [12,13], tocodienols (T2s) and tocomonoenols (T1s) [14,15,16,17]. Research on the latter two has been gaining momentum in the past decade, and new commercial standards of tocochromanols or their metabolites are released to the market every year. An additional challenge is faulty choice of analytical tools and/or chromatographic conditions, which may ultimately lead to the misidentification of tocochromanols in a difficult matrix, for instance, in roasted samples [18]. The possibility of tocochromanols overlapping with the new compounds formed during roasting should be excluded and verified by using mass spectrometry in order to correctly interpret the results [19]. The determination of some tocochromanols is challenging due to their relatively low concentration in plant, food, and biological matrices, especially T1s and T2s; therefore, the choice of an appropriate analytical tool is crucial. At present time, ‘green’ technologies and economic issues are becoming more important in each aspect of our lives, including analytical chemistry. Hexane, a harmful solvent, is still one of the most often used for tocochromanol determination, starting from extraction and ending with determination [20,21]. Therefore, to avoid hexane, heptane has been proposed as a less toxic option [22], which is a solution that is still not satisfactory enough from a health and environmental point of view. In the present review, we set ourselves the goal of compiling a compendium of knowledge on the presence of various tocochromanols in plants and their detection by the application of optimal tools, focusing on important issues for the correct identification and interpretation of the obtained results and appropriately used terminology. This paper discusses the different tools, methods, and techniques for the determination of major (Ts, T3s, and PC-8), minor (T2s and T1s), and less described esterified tocochromanols. The issue of extractable and non-extractable tocochromanols, methods for obtaining analytical standards of tocochromanols, and environmentally friendlier approaches were also discussed. Less common tocochromanol-related compounds not described in the present study, such as plastoquinones, ubiquinones, tocochromanol acids, and other related compounds, have already been reported in detail in previous papers [5,23].

## 2. Tocochomanols—History, Terminology, Structure, and the Main Known Sources

### 2.1. Tocopherols (Ts), Tocotrienols (T3s), and Plastochromanol-8 (PC-8)

Ts and T3s are probably the most recognized tocochromanols. From the chemical perspective, all tocochromanol compounds are similar in structure—these compounds share the chromanol moiety and a side-chain.

Ts, apart from other tocochromanols, are unique with full saturation of the side chain (Figure 1).

As Ts have three chiral centers, a total of eight stereoisomers for each T are possible (RRR, RSR, RRS, RSS, SRR, SSR, SRS and SSS), resulting in a total of 32 possible T stereoisomers [24]. In nature, α-T can be found as RRR- stereoisomer, which is considered to have the highest bioavailability, while commercial standards and dietary supplements of α-T or some other Ts or T3s are often synthesized by non-stereospecific reactions, and these standards include a racemic mixture of all stereoisomers (rac-α-T) [25]. The side-chain of T3s contains three unsaturated carbons at positions 3′, 7′ and 11′, and therefore, T3s have only one chiral center and only two possible stereoisomers—R and S (Figure 2).

However, combining both stereoisomers and possible combinations of geometrical (cis/trans-) isomers, a total of eight stereoisomers per one T3 homologue is possible [24]. Four major homologues (α, β, γ, δ) of Ts and T3s can be distinguished, which are differentiated by the number of methyl substituents as well as the substitution place in the chromanol moiety of the molecule. There are three methyl substituents in the chromanol ring for homologue α-(5,7,8-trimethyl-), two methyl substituents for β-(5,8-dimethyl-) and γ- (7,8-dimethyl-) and one methyl substituent for δ-(8-methyl-). A compound without methyl substituents but possessing one hydroxyl group (6-hydroxy-chromanol-) and the inherent T side-chain is trivially called a tocol but if it has a T3 side-chain—desmethyl-T3 [23]. Another compound is didesmethyl-T3, which is a T3 without a methyl group in position R4 (Figure 2). Both desmethyl-T3 and didesmethyl-T3 have been found in rice bran [26]. It can be calculated that there could be a total of eight different 6-hydroxy-chromanol-related compounds differentiated by the number and placement of methyl substituents—one without methyl substituents (tocol), three mono-methyl, three dimethyl, and one trimethyl (α-) (Figure 1). Different authors use Greek letters to mark homologues other than α, β, γ, and δ [23,24,27]; therefore, in the future tocochromanol homologue nomenclature should be systematized, especially if the occurrence of the compound can be confirmed in nature. 5,7-Dimethyltocol [24], tocol [28], and α-tocopheryl acetate (α-T-Ac) (Figure 3) [29] are often used as internal standards because these compounds meet the main criteria of being structurally close to the analytes and are not included in biosynthetic pathways of tocochromanol formation in photosynthetic organisms (main matrix of interest).

The high concentration of individual Ts and T3s with their dominance can be found in nature (Figure 4 and Figure 5, Appendix A).

α-T is the main tocochromanol found in leaves of different species [30,31,32,33] and seed oils, such as safflower, cultivated (*Carthamus tinctorius*) and wild (*Carthamus oxyacantha*) [34], almonds (*Prunus dulcis*), Guelder rose (*Viburnum opulus*), Sea buckthorn (*Hippophae rhamnoides*), milk thistle (*Silybum marianum*), wheat germ (*Triticum aestivum*), sunflower (*Helianthus annuus*) [12], and Japanese quince (*Chaenomeles japonica*) [35]. β-T is often stated as a minor tocochromanol or does not occur at all in seeds and plant oils [36,37]. However, not only the presence but even β-T dominance over other tocochromanols has been clearly shown in several reports on different plant materials, for instance, wheat germ (*Triticum aestivum*) [38], robusta and arabica coffee beans (*Coffea canephora* and *Coffea arabica*) [19,39], oak acorns (*Quercus rubra*) and oil thereof [12,40], apple (*Malus* spp.) seeds and seed oils [41,42], Guelder-rose (*Viburnum opulus*) seeds and seed oil [12,43], and kirkir seed oil (*Vangueria madagascariensis*) which was initially reported as kerkir seeds (*Catunaregam nilotica*) [44], but later re-identified as kirkir *V. madagascariensis* [45]. γ-T is the main T in most seed oils [35,36]. The highest concentration of γ-T can be found in corn oil (*Zea mays*) [36], canola (*Brassica napus*), chia (*Salvia hispanica*), flax (*Linum usitatissimum*), golden flax (*Linum flavum*), hemp (*Cannabis sativa*), and pumpkin (*Cucurbita pepo*) seed oils [12], as well as in less widespread seed oils, such as red currant (*Ribes rubrum*), pomegranate (*Punica granatum*), and watermelon (*Citrullus lanatus*) [35]. To the best of our knowledge, the predominance of δ-T has only been found in the seeds and seed oils of plants belonging to the Boraginaceae family in the *Borago* genus (*B. morisiana*, *B. officinalis*, *B. pygmaea*, *B. longifolia*, and *B. trabutii*) and *Echium gentianoides* [46]. Relatively high concentration of δ-T can also be found in such seed oils as apple (*Malus* spp.) [41,42], red currant (*R. rubrum*) [35], European beech (*Fagus sylvatica*) [47], Guelder rose (*V. opulus*), and soybean (*Glycine* Willd.) [12] (Figure 4, Appendix A).

The occurrence of T3s in nature is much rarer than Ts and is generally limited to non-photosynthetic organs, most notably in monocots [48]. The predominance of α-T3 over other tocochromanols has been observed in both groups of plants—monocotyledon oat (*Avena sativa*) and rye (*Secale cereale*) bran oils [38]—and dicotyledon in seed oils of species belonging to the Apiaceae family (Umbelliferae), such as hemlock (*Conium maculatum*) [49], cumin (*Cuminum cyminum*), and garden angelica (*Angelica archangelica*) [50]. Some authors introduce doubtful or incorrect information on the tocochromanol composition in plant material, which is often based on the limited number of investigated samples, while a faulty choice of analytical tools and wrong interpretation of the obtained results may be another reason for misleading conclusions. For example, [37] paper may state that β-T3 does not occur in plant oils; however, oils obtained from the bran of such monocotyledon plants as spelt (*Triticum spelta*) and wheat (*Triticum aestivum*) are particularly rich and dominated in β-T3 [38]. One example of a dicotyledon plant source dominated by β-T3 is black caraway (*Nigella sativa*) seeds and seed oil [12,51]. The occurrence of β-T3 has also been reported in other cereals, such as barley (*Hordeum vulgare*) [11], oat (*A. sativa*), and rye (*S. cereale*), with the highest concentration in the bran oils [38], and palm oil [12,52]. γ-T3 is probably the most widely distributed T3. Two of the most important sources of γ-T3 are latex from rubber tree (*Hevea brasiliensis*) [53], discovered in 1965, and annatto (*Bixa orellana*) seeds [54]. Annatto seed oil is the richest source of γ-T3 with a concentration up to 2 g/100 g oil [54]. γ-T3 is the main tocochromanol in both monocots, e.g., in oils obtained from species belonging to the Arecaceae family, such as California palm (*Washingtonia filifera*) [55] and oil palm (*Elaeis guineensis*) (unrefined and refined oils) [12]; and eudicots, e.g., berry seed oil of species belonging to the Ericaceae family, such as wild bilberries (*Vaccinium myrtillus*), lingonberries (*Vaccinium vitis-idaea*), Arctic cranberries (*Vaccinium oxycoccos*), crowberries (*Empetrum nigrum*), cranberries (*Vaccinium macrocarpon*) [56], Apiaceae family, such as *Prangos platychlaera*, *Prangos uechtritzii*, *Heracleum platytaenium*, *Heracleum trachyloma*, *Heracleum crenatifolium* [49], coriander (*Coriandrum sativum*), parsley (*Petroselinum sativum*), celery (*Apium graveolens*), dill (*Anethum graveolens*), carrot (*Daucus carota*), caraway (*Carum carvi*), fennel (*Foeniculum vulgare*), and giant hogweed (*Heracleum mantegazzianum*) [50]; and grape seeds and seed oils, regardless of the species, cross and cultivar [57,58,59] with some exemptions, which can be associated with the varying tocochromanol composition at different stages of grape seed development [60]. δ-T3 and β-T3 are less common T3s in nature than α-T3 and γ-T3, which are found mostly in low concentrations. Annatto (*B. orellana*) seeds are an exception as they may contain up to nearly 15 g of δ-T3 in 100 g of oil [54]. Up until 1965, palm oil was the only known source of δ-T3. In 1965, δ-T3 was found in rubber tree latex (*H. brasiliensis*) [53]. δ-T3 can be also found in seed oil from crowberries (*E. nigrum*) [56], cranberries (*V. macrocarpon*) [12,56] and in several species belonging to the Apiaceae family [49,50]. In all of the listed sources of δ-T3, the concentration of δ-T3 is 1000 times lower in comparison to annatto seed oil. The composition and concentration of tocochromanols in the plant material depends not only on the species but also plant vegetative part [61], genotype [62], stage of development [31,60], abiotic factors during plant growth [62], and sex of some plants, e.g., sea buckthorn (*Hippophae rhamnoides)* [30] (Figure 5, Appendix A).

PC-8 was first identified by Whittle et al. [64] in the leaves of *Hevea brassiliensis*. PC-8 is also named γ-toco-octaenol and is similar to γ-T3, except γ-T3 has three unsaturated isoprene units in the side-chain, while PC-8 has eight unsaturated isoprene units (Figure 6).

PC-8 is found in the seeds and oils of various species belonging to different families [36,63,65]. The oils richest in PC-8 are *Linum usitatissimum* [65], which is often used for PC-8 isolation [12,13], and *Erythrophleum fordii* [63]. PC-8 is present in different plant organs [61,66], especially leaves of both eudicots, e.g., *Pseudobombax munguba* [65], and monocots, e.g., *Zea mays* [67]. The content of PC-8 in leaves depends on the species and the stage of development [30,61,66,68] (Figure 7, Appendix A).

### 2.2. Tocomonoenols (T1s) and Tocodienols (T2s)

T1s and T2s are less common in nature and can be found in lower concentration than Ts and T3s [14,16]. 11′-α-T1 was first isolated and purified from palm and rice bran oils in 1995 [69]. Relying on the evidence of double bond location in T3s, it can be assumed that in T1s, the double bond can be located on the 3′, 7′ or 11′ positions. However, a novel 12′-α-T1 was discovered in the eggs of *Oncorhynchus keta* in 1999; it was named “marine-derived tocopherol” (MDT) [70] and later was found in other marine organisms [71] (Figure 8).

Apart from this, in studies on T1s in plants with emphasis on thorough structure identification, only 11′-T1s have been structurally confirmed, and in cases where precise identification was not possible, detected T1s were assumed to be 11′-T1s (Figure 8). T1s were identified in palm oil (11′-α-T1) [72], T3-rich fraction of palm oil (11′-α-T1) [52], countercurrent chromatography (CCC) fraction of palm oil (11′-α-, 11′-β-, and 11′-γ-T1) [73], in pumpkin seed oil and its CCC fraction (11′-α-, 11′-β-, 11′-γ-, and 11′-δ-T1) [16], leaves of *Kalanchoe daigremontiana* and *Phaseolus coccineus* (11′-β-, 11′-γ-, and 11′-δ-T1) [74], kiwi (*Actinidia chinensis*) peel and pulp (11′-δ-T1) [75], in cyanobacteria and microalgae (11′-α-T1 and MDT) [15] (Figure 9). By analogy with T3s, T2s theoretically can have two unsaturated carbons at positions 3′ and 7′, 3′ and 11′ or 7′ and 11′ (Figure 6). α-T2 was identified in 1996 in the palm oil [76], and later, T2s were identified in the T3-rich fraction of palm oil (7′,11′-α-T2) [52], in CCC fraction of palm oil (3′,11′-α-T2 and 7′,11′-α-T2) [73], and in pumpkin seed oil and its CCC fraction (3′,11′-γ-T2, 7′,11′-γ-T2, 3′,11′-α-T2 and 7′,11′-α-T2) [16]. Lastly, it was proposed to call 3′,11′ T2s, 7′,11′ T2s and 11′-T1s ‘tocoflexols’ due to an increased side-chain flexibility in comparison with T3s [77,78] (Figure 10).

### 2.3. Bound Tocochromanols

The existence and physiological functions of certain T esters was first raised in 1943 [79]. Acetate, succinate, and phosphate α-T esters are vitamin E (α-T) precursors with more polar characteristics than α-T (a non-polar molecule), and therefore, they show better solubility in water. Generally, precursors of vitamin E (α-T) are synthesized for commercial purposes, e.g., supplements due to their higher resistance against oxidation (increased shelf-life) compared to free tocochromanols [80]. Along with recent ones, reports from as early as 1960s and 1980s show that bound tocochromanols are also present in nature. However, few studies are dedicated to this specific topic.

Tocochromanols in the form of esters (Figure 11) are present in such oils as rice bran oil (7% total, 5% α-T esters), soybean oil (1% total, γ-T and δ-T esters 0.5% of each), sesame oil (1% α-T esters) [81], and palm oil (4–14% total, 3–11% α-T esters) [82]. Esters of tocochromanols are particularly abundant in the lipid fraction of latex obtained from rubber trees (*H. brasiliensis*) and constitute 45–57% total tocochromanols, depending on the applied hydrolysis (acid and alkali), where γ-T3 is the major ester [53]. Bound Ts have also been found in rape, soybean, and flax seeds, where rapeseeds were characterized by the highest amount of bound Ts (67%) [83] and hazelnut (*Corylus avellana*) (22–46%) [84]. A varied composition and concentrations of T esters can be found in bell pepper, chili pepper, cucumber, walnut (7–84%), where bell and chili peppers are dominated by α-T esters, while walnut and cucumber are abundant in δ-T and γ-T esters, respectively [10]. In cucumber, walnut, and bell and chili peppers, the following esters have been found: α-T with C12:0, C14:0, C15:0, C16:0, and C16:1Δ9; β-T with C12:0; γ-T with C12:0, C15:0, C14:0, C16:0, C18:1∆9, C18:3∆9,12,15, and C20:0; and δ-T with C12:0, C14:0, C16:0, C18:0, and C20:0 [10]. The esters of α-T, β-T, or γ-T with such saturated fatty acids as C12:0, C14:0, C16:0, and C18:0 have been found in leaves and flowers of *Nuphar luteum* and stems and leaves of *Nymphea alba* [85]. The concentration of esters in investigated samples is affected by the plant maturity [86] and cultivar (genetic factor) [84,86]. Tocopheryl fatty acid esters can be found in nature [10], while also can be formed as side products, e.g., during oil heating at frying temperature [87] and oil deodorization [88]. Commercially deodorized oils such as rapeseed and sunflower can contain 3–12 mg/kg of α-T esters [88]. Fatty acid esters can also be found in the purified residues and distillates of structured lipids [89]. Additionally, bound tocochromanols may occur as glycosides and can be obtained, e.g., by the biotransformation of Ts into glycosides by cultured plant cells of *Eucalyptus perriniana* [90]. Nevertheless, tocopheryl fatty acid esters, in contrast to α-T-Ac, are characterized by a lower bioavailability than free α-T, and they most likely are non-fissile in the human body [10]. Hence, it is important to determine free and esterified tocochromanol separately in future studies on the esterified tocochromanols and other potential biological activity [10].

## 3. Extraction of Tocochromanols

### 3.1. Sample Preparation and Extraction Techniques

The sample preparation method used before analysis is one of the main factors affecting the uncertainty of the analysis results. The choice of a correct sample preparation technique depends on several factors, the main ones being the purpose of the study, detection technique, amount of analytes in the sample matrix, and the type of the sample. Such aspects as the extractability, free and bound tocochromanols, and stability of the analytes should be taken into consideration. This section describes the main sample preparation techniques used in the analysis of tocochromanols as well as the main issues associated with extraction.


*Sample Dilution*


For several types of samples rich in fat, such as plant oils [12,35], butter [91], and margarine [92], sample dilution in 2-propanol prior to the analysis of Ts and T3s by RPLC has been proposed. The fat samples were proposed to be diluted also in other solvents, such as *n*-hexane [93], cyclohexane [94], or methyl tertiary-butyl ether:methanol (1:1 *v/v*) [95]. The type of used solvent for sample dilution and its proportion is affected by several factors, such as the sensitivity and selectivity of used detection method, concentration of tocochromanols in the sample, technique used for their determination (e.g., *n*-hexane is suitable for NPLC, while 2-propanol for RPLC), analytical tools (columns) [12], and type of fat sample (e.g., butter and margarine cannot be dissolved in *n*-hexane due presence of water but can be dissolved in 2-propanol) [91,92]. Due to concerns about tocochromanol stability, some authors prefer to add antioxidants such as butylated hydroxytoluene (BHT) to the sample and use amber glass to protect analytes against oxidation [94]. It should be taken into account that since the sample is not purified, simple dilution technique may lead to the misidentification and misinterpretation of the results, especially if the analyzed matrix, such as cold-pressed oils, is poorly investigated and is rich in many compounds. Sample dilution is not necessarily compatible with all chromatographic techniques due to possible contamination concerns; however, some authors adopt this method for T and T3 detection in oil samples using the RPLC-MS [96] and SFC-MS systems [94].


*Direct Solvent Extraction*


Direct solvent extraction (solid–liquid extraction) is a method used for powdered samples which require the extraction of tocochromanols from the matrix. Briefly, the powdered plant material is supplemented with different solvents, such as methanol, or *n*-hexane:ethyl acetate (9:1 *v*/*v*) [11], 80% ethanol [30], *n*-hexane [97], ethanol [98,99], and some other solvents in various ratios to plant material, with the addition of antioxidants such as BHT [97,98,99] or pyrogallol [11] and extracted only by mixing [11,97] or treated with ultrasound [30,98,99] at room or elevated temperatures. The application of various solvents for the direct extraction of tocochromanols has been deeply discussed and reviewed previously [100]. In the case of samples with a high enzyme activity, stabilization is advised to prevent tocochromanol destruction [101]. The main flaw in most of those studies is the lack of comparative study investigating extractability when using other solvents and extraction techniques, e.g., saponification, and method validation. Saponification protocol is stated as the most effective in tocochromanol extraction [11,30,102]. The advantages of direct extraction are speed, low workload and costs; it is an environmentally friendlier sample preparation technique if ethanol is used and compatible with NPLC or RPLC when hexane or ethanol and methanol are applied, respectively. The disadvantages are lower extractability and often the presence of other analytes in the extract, e.g., phenolic compounds or lipids (type of used solvent is a key factor), which can negatively affect identification depending on the method of detection.


*Saponification*


Saponification (alkaline hydrolysis) is by far the most common protocol for sample preparation for tocochromanol determination in different types of matrix [11,30,102,103,104]. The aim of this procedure is to improve the extractability of tocochromanols (liberating tocochromanols from ester forms and the matrix) and partly purify the sample (prevents the emuslification, the presence of sample particles, and the high amount of lipidic compounds) [11]. Tocochromanol assays obtained by saponification protocol show the total content of free and bound tocochromanols. In the process of saponification, generally, solutions of potassium hydroxide (KOH) with a concentration from 5 to 20% [83] up to 80% are used, with 60%, *w*/*v*, being the most effective [105]. Prior to adding the KOH solution, the sample is supplemented with ethanol and different antioxidants (BHT, pyrogallol, and/or ascorbic acid) to prevent the destruction of tocochromanols during the process of saponification. The reaction takes 10–120 min [83], with 25–45 min applied in most reports [11,30,102,103] at ambient temperature [106] or up to 100 °C [102]; however, the range 70–80 °C is most commonly used [11,30,103]. During saponification, the sample is mixed every 5–10 min [11,30,103] or treated with ultrasound [104]. Then, 1%, w/v, sodium chloride (NaCl) is often added to end the reaction and to reduce the surface tension between the phases [11,30,103]. The sample is cooled in an ice bath and, finally, tocochromanols are extracted by *n*-hexane [106] or a mixture of *n*-hexane:ethyl acetate (9:1 or 8:2, *v*/*v*) [11,30,102,103]. A 9:1, *v*/*v*, ratio of *n*-hexane to ethyl acetate is most often used [11,30,103]. After multiple re-extractions, the sample is evaporated and dissolved in the suitable solvent for the specific detection method. The saponification protocol is the most effective tocochromanol extraction method [11,30,102]. However, during the process, the sample is firstly saponified and only then extracted; therefore, we do not know whether obtaining a higher content of tocochromanols is the result of liberating bound tocochromanol forms or improving extractability, or both. There is no proof that the saponification protocol provides 100% tocochromanol recovery from the sample matrix. For instance, it has been shown that thermal (roasting) or chemical (decaffeination) sample treatment before saponification improves tocochromanol extractability from coffee beans [19].


*Alternative Techniques*


Soxhlet extraction is the most often method used for oil extraction [38,107], but it is only occasionally used for tocochromanol analysis in comparison to other sample preparation methods [108]. Soxhlet is less effective in tocochromanol extraction from cereal matrixes than saponification [108], and the content of tocochromanols is affected by the type of used solvent in Soxhlet apparatus [107]. Ultrasound treatment seems to be more effective in tocochromanol extraction than Soxhlet when the same type of solvent is used [107]. In another report, two factorial analysis of variance showed a lack of statistically significant differences in tocochromanol profile and content in oils extracted from Japanese quince seeds by Soxhlet, supercritical fluid extraction (SFE), ultrasound treatment, and cold-pressing [109]. The disadvantages of the Soxhlet method are the long time and large volume of harmful solvents, e.g., *n*-hexane. A microwave-assisted T extraction technique with ethanol and 50% KOH (*w*/*v*) [110] has also been applied; however, it was not compared with saponification protocol. Therefore, it is unknown whether this method is more effective. SFE is a modern and environmentally friendlier technique due to the use of liquid CO_2_. It was highlighted that SFE can be effective in the extraction of tocochromanols (84–100%) from oil crops [111]. However, the SFE method appears to be ineffective for the extraction of tocochromanols from cereal matrixes [108]. From a technical point of view, nearly all SFE systems are designed for the preparation of extracts for commercial rather than for analytical purposes.

### 3.2. Important Aspects in Extraction of Tocochromanols


*Bound and Free Tocochromanols*


The topic of bound and free tocochromanols was highlighted for the first time in 1965 in latex from the rubber tree (*H. brasiliensis*) [53]. Esterified tocochromanols (ETs) have been separated from free tocochromanols in latex by thin-layer chromatography (TLC) [53]. A similar procedure has been applied for rice bran, sesame, soybean, and palm oils [81,82]. Then, esters were saponified and liberated tocochromanols were determined. Both free tocochromanols and tocochromanols liberated during alkaline and/or acid hydrolysis of esters were determined by measuring the absorption spectrum [53] and NPLC [81,82]. It was demonstrated that the liberation effectivity of tocochromanol homologues depends on the type of chosen hydrolysis method (alkaline or acidic) [53]. In various studies different mixtures of solvents, techniques and plant material for the extraction of ETs were used; chloroform:methanol (2:1, *v*/*v*) by the Folch method from rubber tree latex (*H. brasiliensis*) [53], acetone and then dichloromethane by ultrasonication in different plant organs of *N. alba* and *N. luteum* [85], and a mixture of cyclohexane/ethyl acetate (46:54, *w*/*w*) by focused open-vessel microwave-assisted extraction in various vegetables and walnuts [10]. Since none of the applied approaches have been compared, it is difficult to discuss their effectiveness. Future comparison studies are required. Additionally, most achievements in the topic of bound Ts in specific plant materials were investigated by one scientific group that found relatively high concentrations of bound Ts in relation to free Ts, e.g., in rapeseed [83] and hazelnuts [84], which is at least worth re-examination. Therefore, future development of suitable analysis protocols for ETs determination is required. Most of the analysis protocols use saponification followed by the extraction of total tocochromanols (free and liberated from bound forms). It might be one of the reasons for the limited information about extractable and non-extractable tocochromanols. The limited knowledge on the presence of bound tocochromanols in plant material is mainly a result of several challenges: lack of standards and suitable protocols for their determination and the time-consuming performance of such analyses.


*Non-Extractable Tocochromanols*


It has been shown that neither Soxhlet nor the SFE method is effective enough to extract the tocochromanols strongly bound to other components in the cereal matrix [108]. Therefore, the saponification procedure is advised to liberate those tocochromanols [108]. The lower recovery from cereal samples, e.g., barley, can be associated with the presence of ETs or tocochromanols that are bound to the matrix [11]. On the other hand, even the saponification protocol can be insufficient to extract all tocochromanols from plant matrix. It has been shown that the extraction of Ts is slightly improved from coffee beans after such treatments as decaffeination and roasting before the saponification protocol (data supported by RP-UPLC-ESI/MS^3^), probably due to the cellular structure damage, which allows for the extraction of all Ts [19]. The question is: what can we state in such cases about the presence of the non-extractable tocochromanols? The type of used hydrolysis (alkaline or acid) has been known to affect the final result of tocochromanols since 1965 [53]. The terms ‘*non-extractable tocochromanols*’, ‘*non-extractable tocopherols*’, and ‘*non-extractable tocotrienols*’ are not present in scientific literature. Meanwhile, the term ‘non-extractable polyphenols’ has become more and more discussed in the last decade, especially in studies on fruit peels and by-products [112,113,114,115,116,117], and protocols have already been proposed for their determination in different plant materials [118]. Free tocochromanols, due to the presence of one phenol group in the chromanol ring, are also called monophenols [119]. The occurrence of extractable and non-extractable forms would be another common feature of these bioactive compounds. However, such terms and protocols that exist for polyphenols have never been proposed/investigated for tocochromanols. The esterified forms of tocochromanols and non-extractable tocochromanols are high molecular weight molecules. They can be associated to polysaccharides, proteins, and constituents of dietary fiber similarly to non-extractable polyphenols. Additionally, they can be retained in the matrix and inaccessible to solvents due to various interactions with the plant matrix, as is the case of non-extractable polyphenols [118].


*Roasting—Challenges after Sample Treatment for Tocochromanol Determination*


The analysis of roasted samples should be conducted thoughtfully due to newly formed compounds, which are generally known as Maillard reaction products [120]. A majority of compounds formed during the roasting of mustard seeds were found to be non-polar, probably formed from phospholipids, and some of them may exhibit fluorescence properties [120]. Since a fluorescence detector is one of the most popular detection methods applied for tocochromanol determination, newly formed compounds may interfere with identified tocochromanols and ultimately lead to misinterpretation of the data. While the slight increscent of tocochromanol extraction efficiency from the plant material after roasting, e.g., in coffee [19] or pumpkin seeds [121], can be explained by improved extractability due to plant cell damage, providing an evidence-based explanation for a many-fold concentration increase is difficult. For instance, in thirty two samples of arabica and robusta coffee beans before and after roasting, on average, a nearly four times higher total content of Ts and over thirty-six times higher content of γ-T was observed in the roasted beans [18]. The reason for the obtained results is most likely the coelution of analyzed Ts with newly formed compounds. The unidentified compounds formed during roasting have been observed in several coffee studies during T identification [19,39]. Our thesis, that roasting causes some analytical challenges during tocochromanol determination by using fluorescence detection especially in NPLC, is best supported by three reports on roasted rape seeds. An increased amount of tocochromanols during roasting was observed in all three studies. The three reports generally support each other if total tocochromanols are considered; however, different effects on individual tocochromanol content were observed. The content of γ-T [122], β-, δ-T [123], and PC-8 [124] increased in roasted rapeseeds. In each of those reports, a NPLC-FLD system with different columns and mobile phase modifiers was used. This phenomenon of the differing effect on tocochromanol content during rapeseed roasting requires future investigation using other analytical tools for tocochromanol determination, for instance, RPLC, SFC, gas chromatography (GC) or application of MS detection.

### 3.3. Lab-Scale Isolation, Synthesis, and Purification of Tocochromanols

One of the challenges in the detailed characterization of tocochromanols in plant material is associated with the lack of commercially available standards (e.g., PC-8, T1s, T2s, and tocochromanol esters). Thus, sometimes, research on rarer tocochromanols is not possible without including isolation and purification methods as well as synthetic approaches to obtain products as qualitative or quantitative standards. In addition, another reason for the acquiring of tocochromanols is to further use the products in studies on biological activity. There are few approaches for the isolation of specific tocochromanols.

To the best of our knowledge, the main criteria for choosing the specific plant material for the isolation of tocochromanols are a high concentration predominance of the compound over other tocochromanols. The right plant material and purification techniques could increase the final purity of the isolate. Plant material with dominance and high concentration of specific T and T3 have been listed in Section 2.1. For instance, PC-8 has been isolated from saponified flaxseed oil (containing mostly γ-T and PC-8) using NPLC with a semi-preparative column [13]. The chromatographical purity of PC-8 was tested using LC with a diode-array detector (DAD), and the final purity of the isolate was 93 %. The concentration was measured gravimetrically. The concentration of tocochromanols such as PC-8, Ts, and T3s can be also measured spectrophotometrically using molar absorption coefficients [13,125,126]. The molar absorption coefficient is affected by the type of used solvent for measurement [13]; however, in the case of other minor tocochromanols, molar absorption coefficients are not available. A similar approach as reported by Siger et al. [13] was used to isolate α-, β-, γ-, δ-T3s and PC-8 from different plant oils by applying SFC with an analytical-scale column [12]. SFC was also proposed to isolate α- and β-T from wheat germ oil using a semi-preparative column [127]. Re-chromatography [13], low-pressure liquid chromatography [128], column chromatography and TLC [129] have been applied to purify PC-8. Similar methods can be used to increase the purity of other tocochromanols. For the isolation of α-T1 and MDT [130] from tuna oil, first, the sample was saponified; later, column chromatography (using silica and *n*-hexane:ethyl acetate 95:5 *v*/*v*) and two fractionations using the HPLC system were applied. Isolation of a novel 11′-δ-T1 homologue from kiwi fruits has been performed by extraction with *n*-hexane; subsequently, column chromatography (silica—petroleum ether/acetone mixture and NH_2_-silica—petroleum ether/ethyl acetate mixture) and RPLC preparative chromatography (RP-8 column—methanol:acetonitrile:H_2_O mixture) were used to fractionate the extract [75]. Counter-current chromatography (CCC) is the latest trend in fraction enrichment with T1s and T2s [16,73]. Higher sample amounts can be loaded into the system due to the absence of the stationary phase; this advantage makes CCC beneficial for the fractionation or isolation of compounds with low content in the plant material, including tocochromanols.

A synthetical approach is another alternative to obtain tocochromanol-related standards. For example, PC-8 has been obtained from plastoquinone-9 [64]. α-T1 and α-T2 have been obtained by saturating double bonds in the side-chain of α-T3 using a partial hydrogenation reaction with palladium-activated carbon ethylenediamine complex as a catalyst [131]. (2R,8′ S,3′ E)-δ-T2 has been obtained in an eight-step synthetic route from δ-T3 [77]. Different α- and γ-tocopheryl fatty acid esters have been obtained by esterification reaction with the subsequent separation of free fatty acid as well as unreacted free T from the mixture using solid phase extraction [10]. α-Tocotrienolquinone epoxides were obtained by the thermal oxidation (3 h, 50 °C) of α-T3 with the addition of azobisisobutyronitrile in oxygen-saturated acetonitrile/water (60:40 *v*/*v*) mixture, while 5-formyl-γ-T3 was obtained as a γ-T3 thermal oxidation product (60 °C for 3 h) [132].

## 4. Chromatographic Techniques Used for Tocochromanol Determination

### 4.1. Liquid Chromatography (LC)

LC, including normal-phase (NP) and reversed-phase (RP), is the most common technique for T and T3 determination. Using phrases “tocopherol + normal phase” and “tocopherol + reverse phase” in the www.scholar.google.com (on 14 May 2021) search engine, 150,000 and 61,800 results are returned, respectively. While it is not necessarily the most precise way to evaluate the trend, it can be generalized that NP was applied more often than RP. One of the reasons for this predominance of NP over RP in the analysis of Ts could be that NP was introduced earlier than RP.


*Normal-Phase (NP)*


The separation of T homologues (α, β, γ, and δ) in oils was performed for the first time in 1973 using NP on silica columns (Corasil II) by applying hexane:diisopropyl ether (19:1, *v*/*v*) as the mobile phase and fluorescence detection with an λ_ex._ of 295 nm and an λ_em._ of 340 nm [133]. In NPLC, *n*-hexane or *n*-heptane [13,22,52] are generally used as the mobile phase with different modifiers: 2-propanol, dioxane, ethyl acetate, tetrahydrofuran, methanol, tert-butyl methyl ether, diisopropyl ether, and acetic acid up to 5.5% [22,134]. Generally, three main different stationary phases are used for NPLC: diol, amino (NH_2_) and silica (Si), the latter one being the most popular NP stationary phase for tocochromanol determination [134]. Usually, separation is performed at elevated temperatures; this parameter as well as the flow rate, type and concentration of the modifier are adjusted for the specific column to achieve the best ratio between separation selectivity and total analysis run time [29,134]. In NPLC, the number and position of the methyl substituents on the chromanol ring has higher impact on the compound polarity and thus affinity than the saturation and length of the side chain. It is more apparent when PC-8 is separated simultaneously with Ts and T3s [13]. The highest difference in polarity prevails between homologues α (less polar) and δ (more polar), which have in total three (5,7,8-trimethyl-) and one methyl (8-methyl-) substituents on the chromanol ring, respectively. Therefore, this pair is not considered problematic to separate. On the other hand, β and γ isomers have two methyl substituents on different positions (5,8-dimethyl and 7,8-dimethyl, respectively), and the separation of this pair is considered to be problematic, as structural asymmetry is sometimes not enough for successful separation. The difference in polarity between Ts (less polar) and T3s (more polar) arises due to the saturation degree of the side chain. To sum up, the typical elution order of tocochromanol homologues on the NPLC Si phase is α-T, α-T3, β-T, γ-T, β-T3, γ-T3, δ-T, δ-T3 [134]. α-T-Ac is commonly used as an internal standard and will elute first, since it has reduced affinity to the stationary phase due to the absence of the hydroxyl group on the chromanol ring [29], while 5,7-dimethyltocol is also a viable option to use as internal standard [135]. Diol is another NP stationary phase widely used for tocochromanol separation with the same elution order as Si columns, but it is less reproducible, stable and efficient than Si phases [29,134]. A valuable alternative for the NP separation of tocochromanols is the NH_2_ phase, which can be used in NP as well as in RP mode. The NH_2_ phase in NP mode may provide an identical elution order as the Si and diol phases [29,134] or a slightly different order to give α-T, α-T3, β-T, β-T3, γ-T, γ-T3, δ-T, and δ-T3 [134]. Known advantages of NPLC for the determination of tocochromanols are high selectivity, especially for the separation of isomers β and γ, the ability to simultaneously separate PC-8 from Ts and T3s [24,136], compatibility of the mobile phase with the direct dilution of oils [137], and compatibility with direct analysis of sample diluted in *n*-hexane after saponification [137]. However, the usage of toxic solvents harmful to the environment and the necessity to remove water in instrument channels and switch the instrument for work with NP mode are considered to be the main disadvantages of NPLC.

PC-8 could be considered a γ-tocochromanol homologue but with a longer side chain, however, their biosynthetic pathways are different. Because the retention of tocochromanols in NPLC is more dependent on the differences in the chromanol moiety of the molecule, it is easier to elute/separate it in the NPLC (PC-8 is eluted between β-T and γ-T3) than RPLC [13]. PC-8 is often determined simultaneously with four Ts and four T3s; however, except for two reports with separation run time 40 min [13] and 60 min [136], no papers showing NPLC chromatogram examples have been published [63,93,138]. The lack of separation selectivity and the fact that a PC-8 standard is not commercially available and is rarely specially isolated may become the reasons for misinterpretations of the obtained chromatograms. When the plant material does not contain β-T3 and γ-T3, chromatographic challenges in NPLC are not observed, and PC-8 is eluted immediately after γ-T [139]. On the other hand, when β-T3 and/or γ-T3 occur in plant material, the challenge of PC-8 separation arises, since PC-8 is eluted between this problematic pair. Thus, not only the type and concentration of used modifier for the NP mobile phase but also its purity (data unpublished) are critical. Additionally, PC-8 can also be problematic for the correct identification in the roasted samples due to newly formed compounds and their possible coelution with PC-8. For instance, it has been reported that the PC-8 content can increase more than 26 times in rape seeds during roasting [124]. Such a large PC-8 concentration increase during roasting is difficult to explain other than with the co-elution of PC-8 and compounds formed during roasting. If the results obtained are controversial, we recommend re-examining the samples using additional analytical tools to eliminate misidentification. Appendix A summarizes NPLC chromatographical methods for the determination of tocochromanol-related compounds.


*Reverse-Phase (RP)*


RPLC started to replace NPLC as an alternative method for tocochromanol determination over two decades ago. Unlike NPLC, RPLC utilizes relatively non-polar stationary phases (e.g., C18, C30) and different mixtures of polar eluents. While the basic silica NPLC stationary phase provides better selectivity for tocochromanol separation, advancements in the technology of RP stationary phase preparation allow achieving baseline separation of the analytes. The use of such mobile phases as methanol and acetonitrile improves the sustainability of the chromatographical methods in the context of the environmental desirability and reproducibility of the obtained results by RPLC. Generally, tocochromanols are compounds of lipophilic (non-polar) nature, and the presence of the relatively long side-chain of tocochromanols makes the elution in RPLC demanding for the strong eluents, meaning a high concentration of the organic phase (methanol, acetonitrile) in the mobile phase. In T and T3 separation, the isocratic flow of a mixture of methanol and water in ratio from 80:20 to 100:0, *v*/*v*, is most often used [140,141,142,143] (Appendix A). Generally, the concentration of water in the mobile phase is associated with the expected results of separation and number and type of tocochromanols to be separated. It can also depend on the used column phase and its parameters. The addition of water to the pure organic modifier (e.g., methanol) is positively correlated with peak resolution and negatively correlated with the overall sensitivity of method [144]. Acetonitrile, methanol and water [36] or acetonitrile, methanol and dichloromethane are also used quite often on C18 and C30 columns [43,145]. While the isocratic separation of tocochromanols is the most popular, certain samples may include lipophilic compounds that will not be eluted during the isocratic chromatographical run and will most likely elute during the next run. While the drawbacks of the gradient elution include significantly longer analysis time, some authors prefer using gradient elution utilizing methanol:water (96:4) and methanol:methyl tert-butylether:water (4:92:4) [41], methanol and methanol:2-propanol:acetonitrile (40:50:10) [146], methanol:water (80:20) and methanol [142], methanol:water (91:9) and tert-methylbutylether:methanol:water (80:18:2) [101] solvent systems.

The C18 column is often the first choice for the RPLC separations of most analytes due to its universality and availability in most scientific laboratories, however it does not separate β and γ tocochromanol isomers. There is only one published report on the separation of tocochromanols using a PerfectSil Target ODS-3 (octadecyl functional group-C18) column; however, separation was performed at 7 °C, and the total analysis run was 62 min [147]. The elution order of T3s and Ts separated by a C18 column in RPLC is as follows: δ-T3 **>** β-T3 + γ-T3 **>** α-T3 **>** δ-T **>** β-T + γ-T **>** α-T [36]. The separation of as many tocochromanols as possible is particularly important when the plant material is investigated for the first time. In the last two decades, several types of stationary phases have been developed, which are capable of separating the β and γ isomers. These stationary phases can be grouped in two groups differing by the elution order between the β and γ isomers—the first group is δ > γ > β > α, while in the second group, δ > β > γ > α. The first group includes C30 columns [36,141]; π-naphthalene (π-NAP) columns—the naphthalene bound stationary phase enhances π–π interactions and improves selectivity for structural isomers such as β and γ [141,148]; and pyrenylethyl (5PYE) columns with 2-(1-pyrenyl) ethyl groups bonded silica offer separation with high molecular shape selectivity or π–π interactions providing high selectivity for the separation of structural isomers [149]. The second group includes pentafluorophenyl (PFP) columns with pentafluorophenyl propyl (PFP/F5) ligands bound to the stationary phase [28,91,140,150,151], and selectivity is achieved through four mechanisms of interaction (hydrogen bonding, dipole–dipole interactions, hydrophobic, and π–π interactions), allowing for structural isomer separation [152]; and RP-Amide columns, which contain a base-deactivated phase with a polar group within the alkyl-bound phase, which provides unique selectivity [150]. The first report about the separation of T isomers (β and γ) using a PFP column was published in 1994 [151], and in only two decades, it had become popular for the separation of four T and four T3 homologues [91,101,140].


*PC-8 Issue*


One of the disadvantages of the RP is the inability of simultaneous separation of PC-8 with four Ts and four T3s in one chromatographic run, which is due to the insufficient eluting strength of the RPLC mobile phase (usually methanol or acetonitrile) and strong interaction of the PC-8 side-chain with RP stationary phase. The analytical determination of PC-8 using RPLC with a C18 column can be achieved by the addition of *n*-hexane (up to 20%) to the mobile phase [36] or by decreasing the column length, ID and particles (50 × 2.1 mm, 1.7 μm), increasing the column temperature (60 °C), and using 100% methanol [153]. In both cases, the coelution of most Ts and T3s prevents the effective simultaneous separation of these major compounds [36,153]. Nevertheless, both methods are suitable for the determination of PC-8, but Ts and T3s should be detected with a different appropriate method. Therefore, when tocochromanols are determined in green plant material (especially leaves), seeds, and oils rich in PC-8, NPLC and SFC with an established method for the separation of Ts, T3s, and PC-8 [12] seem to be better options than RPLC, or the use of two separate chromatography runs by RPLC should be performed to obtain detailed information about all tocochromanols in the investigated plant material [36].


*C18 Column Issue*


By using phrases “tocopherol + C18 column”, “tocopherol + C30 column”, and “tocopherol + PFP column” in the www.scholar.google.com (on 14 May 2021) search engine, the following number of the results is returned: 46,900, 4110, and 995, respectively. Apart from the fact that these specific keywords do not necessarily need to be directly related to the description of the methodology for the determination of Ts, it can be assumed that the C18 stationary phase might be the most popular choice for T determination by RPLC. Unfortunately, the C18 column has one serious disadvantage: it is unable to separate isomers (β and γ) [144,146,154]. Due to the lack of selectivity for β and γ separation, the results are often described as a sum of β + γ or, worse, as γ homologue content, while the β presence is ignored. Sometimes, it is stated that β-T is only a minor component in plant material, while β-T3 does not occur at all. Such statements are often based on a limited number of reports, therefore, when the presence of β-T or β-T3 cannot be excluded in the tested sample, appropriate tools for β and γ isomer separation should be used to avoid wrong interpretation of the results. Finally, the presence of β isomer can be expressed incorrectly as γ, e.g., in the seed oils of Guelder rose (*Viburnum opulus*) [56] and Nigella (*Nigella sativa*) [37]. The widespread use of a C18 column instead of other RP stationary phases for tocochromanol separation is not necessarily linked with a lack of knowledge about the most suitable analytical tools but with the universality of C18 columns for several different applications, including tocochromanol determination [155]. The application of the C18 column is often additionally justified by using the statement provided above about very low β homologues occurrence in nature. Recent studies show that the occurrence of β homologues in the plant world is underestimated, which is mainly due to improper methodology, tools, and unsubstantiated assumptions [40]. Such thinking can often lead to misinterpretation, as has been demonstrated in Table 1.

The examples given above on the presence of β-T and β-T3 (Section 2.1) show that the low occurrence of homologue β in the plant world has been incorrectly assumed. It reinforces the importance of using correct tools for tocochromanol determination. The more detailed determination of minor tocochromanols can be a powerful tool, for instance, in the study of authenticity of plant oils [159].


*Detectors Used in LC*


Different studies include several common detectors that have been applied in the LC analysis of tocochromanol-related compounds: diode array detector (DAD), fluorescence detector (FLD), evaporative light scattering detector (ELSD), electrochemical detector (ECD) as well as different mass spectrometric (MS) techniques. ECD is one of the most sensitive and specific detection methods [160,161], while ELSD and UV are non-selective and less sensitive than FLD [162]. ECD is a unique detection method for tocochromanols, allowing to also measure oxidative products of tocochromanols, which do not fluoresce, while UV detection is not sensitive enough for those compounds. ECD is compatible only with RPLC, since only RP eluents are sufficiently polar to carry electrolytes that are required in the eluent solution to support the redox reactions that are the basis for detection [163]. Tocochromanols have fluorescence properties due to the presence of chromanol moiety of the molecule, and the FLD is the most common detector due to its sensitivity and selectivity [11,30,35,102]. While the UV/DAD detector is a viable option for the determination of tocochromanols (at the wavelength around 295 nm) with relatively low sensitivity and selectivity in comparison to FLD (an λ_ex._ of 290 nm and an λ_em._ of 330 nm), it is advised to couple both DAD and FLD to confirm detected analytes, verifying their UV spectra if the concentration allows it [162]. Leray et al. [164] compared the within-run precision of three detectors, and the results were: FLD > UV > ELSD. The sensitivity of the ELSD detector is similar to UV detection; however, it is less selective. Using LC, FLD is around 10^3^ more sensitive in comparison with UV and ELSD, depending on the homologue and conditions [162]. Fluorescence quenching as well as difficulties related to the gradient method development with FLD (baseline is very sensitive to changes in the mobile phase) are the main disadvantages of FLD [141]. Detector sensitivity can generally be lined up as follows: MS ≥ ECD ≥ FLD >> UV = ELSD. Mass spectrum techniques are discussed in Section 4.4.

### 4.2. Gas Chromatography (GC)

GC methods have been used to identify and quantify tocochromanols since the 1970s. However, LC methods are currently more frequently used than GC for major tocochromanols, which can be explained with the longer time of GC analysis, its lower sensitivity, and necessity for a derivatization step [165,166]. On the other hand, GC seems to be the first choice for the determination of minor tocochromanols such as T1s and T2s [142] and bound forms [10,86] due to its better separation selectivity for minor tocochromanols. In GC, a mixture in the gas phase is moving forward in a carrier gas flow (helium, nitrogen, argon or hydrogen) through packed or capillary columns at elevated temperatures. The main factors that affect the results of the analysis are the type of stationary phase and dimensions of the column, carrier gas flow rate, split ratio and injection volume as well as the temperature program and the type of detector used. Tocochromanols are compounds of lipophilic nature; therefore, relatively non-polar (5% phenyl groups) stationary phases are most commonly used for the separation, e.g., HP-5 [167], Rxi-5Sil MS [168] or CP-SIL 8 CB [169]. A cyanopropyl stationary phase CP-Sil88 has been used for the successful separation of α-T enantiomeric pairs (RRS + SSR, RRR + SSS, RSR + SRS, RSS + SRR) [170]. The elution order of Ts and their esters (TMS) is dependent on their boiling point, giving retention time (RT) as follows δ < β < γ < α [171]. While tocochromanols can be separated using isothermal modes depending on the methods’ purpose, matrix and possible interferences, separation is often performed using pre-programmed temperature gradients: 120–350 °C [167], 180–280 °C [168], 150–320 °C [172], 250–290 °C [169], 200–300 °C [173], and 70–310 °C [174]. Such parameters as injection volume, specific split value or splitless mode are dependent on the concentration of tocochromanols in the sample, the purity of the sample, the detection type, as well as the capacity of the column. Nevertheless, splitless [167,168,172] and different split values from 1:5 to 1:60 have been widely applied. Solvents such as ethanol [167,172], *n*-hexane [168], chloroform [174], and *n*-heptane [75] have been used as sample diluents. It is considered that the derivatization procedure is required prior to separation with GC to increase the volatility and thermal stability of the analytes. In some cases, derivatization is required for the improvement in separation selectivity and sensitivity by using MS [171]. While there are some methods that do not require derivatization [167,172], most GC tocochromanol determination methods use trimethylsilyl (TMS) derivatizing agents such as N-methyl-N-(trimethylsilyl)trifluoroacetamide (MSTFA) or N,O-*bis*(trimethylsilyl) trifluoroacetamide (BSTFA) [168,173,174]. Flame ionization detector (FID) is a universal detector for GC that works under the assumption that any gas molecule entering the permanent flame (usually fueled by hydrogen and air) will ionize and cause a change in electric conductivity. However, this type of detector is not selective and does not provide any direct or indirect information on the identity of the compound. This is why tocochromanols are often detected with a MS detector. Hard ionization sources, such as electron impact [75] or softer chemical ionization [173], have been successfully applied for the determination of tocochromanols. The main advantage of the electron impact ionization is the reproducibility of the spectra, which allows for the creation of widely available libraries of mass spectra (NIST Wiley). Therefore, GC-MS is a common tool for the identification of unknown organic compounds [175].

### 4.3. Supercritical Fluid Chromatography (SFC)

In the last decade, there has been a breakthrough for SFC or Ultra-Performance Convergence Chromatography (UPC^2^) systems, which has enabled the acquisition of precision and reproducibility comparable to that of LC systems. Currently, packed column SFC has a meaningful advantage to LC: it uses low-viscosity CO_2_ with modifiers (e.g., methanol, ethanol, 2-propanol) as the main mobile phase, which allows higher operational flow rates, more rapid analysis compared with LC and, additionally, the application of the CO_2_ makes SFC an environmentally friendlier method [176]. The improvement of SFC instrumentation has been demonstrated with recent studies, showcasing the benefits of SFC versus LC as well as their differences in selectivity behavior [177,178,179,180]. The advantages of SFC over LC are the possible application of NP, RP, and HILIC columns while using the same mobile phase in both cases (CO_2_ with a modifier), which provides great versatility in terms of retention mechanisms (NP, RP, and mixed retention mode) [12,181]. Additionally, SFC is characterized by the higher efficiency of molecule separation and rapid analysis time, contributing to the faster method development in comparison to LC. In the last decade, an increased interest can be seen in the application of SFC for tocochromanol determination [182] (Appendix A) using columns deigned specially for SFC [52,182] to core–shell ones used in LC [12,144].

It has been clearly demonstrated that unlike in RPLC, in SFC, the C18 stationary phases provide enough selectivity for the separation of β and γ isomers [12,144,181]. However, it has also been shown that isomer separation is not possible in all C18 columns, and this phenomenon can be a result of different carbon loads of C18 columns by various manufacturers [12]. One of the main disadvantages of the SFC is relative complexity in method development. For example, the elution strength of the mobile phase in SFC is dependent not only on the stationary phase (RP, NP) and the type of modifier (methanol, ethanol, acetonitrile) being used but also on the overall pressure in the system, which is continuously changing depending on the used gradient. It has been demonstrated that the simultaneous separation of PC-8, four T and four T3 homologues (run time 15 min) is possible using the biphenyl bound stationary phase [12]. The obtained elution order was mixed mode: δ-T **>** γ-T **>** β-T **>** α-T **>** δ-T3 **>** γ-T3 **>** β-T3 **>** α-T3 **>** PC-8 [12]. The elution of PC-8 as the last tocochromanol is in agreement with RP order after the addition of a high percentage of *n*-hexane to the methanolic mobile phase of the C18 column [36]. However, in SFC, PC-8 is eluted by the increased concentration of methanol in the mobile phase [12]. Since PC-8 is widespread in plants, especially in leaves [65], it makes sense to use at least flax seed oil to confirm the identification of PC-8 in the future.


*Detectors Used in SFC*


The most commonly used detectors in SFC are UV/DAD and MS with the former used more often. SFC-DAD has been applied for the determination of tocochromanols in plant oil (Ts, T3s, and PC-8) [12] and leaves (Ts and T3s) [183]. The Ts and T3s have absorbance maximums in the UV spectra in the range of 292–298 nm with alcoholic solvents as the mobile phase [144,145], while in the SFC method, which uses CO_2_:methanol (99.8:0.2, *v*/*v*), all homologues have similar maximums in the UV spectra 294–295 nm [144]. The limitation of other detectors, e.g., FLD, in SFC is associated with high pressure in detector measurement cells due to the presence of a back-pressure regulator, which is necessary component to maintain CO_2_ in a liquid stage. The development of a unique cylindrical quartz flow cell is allowed for applying FLD in SFC (up to 20 MPa) [184]. The additional benefits of using FLD in SFC systems, beyond selectivity, are rather minor or limited. The difference in sensitivity between SFC-FLD and SFC-DAD for tocochromanol determination is less pronounced (around 3–20 times, depending on homologue) [184] in comparison to the difference between LC-FLD and LC-UV (about 1000 times) [162]. Application of ECD with SFC has recently been demonstrated for the determination of Ts in vegetable oils [185] and Ts and T3s in nutrition supplements [186].

### 4.4. Mass Spectrometry Techniques

Modern MS techniques allow for the identification and detection of tocochromanols in complex matrixes with high selectivity and sensitivity. Method development, calibration and sample purification is crucial when using MS. This section will briefly cover the main ionization and mass analyzer types that have been previously applied for the analysis of tocochromanols. Bartosińska et al. [171] have published a detailed review paper on MS techniques in the analysis of tocochromanols. The ionization of non-polar substances may be difficult due to the lack of protonation and/or deprotonation sites [187,188]. In order to overcome this problem, it is good practice to use mobile phase additives. The most common approach is the addition of acid (e.g., formic acid and acetic acid) to the mobile phase to increase the formation of protonated species [165]. There are also less conventional techniques. For instance, it has been found that adducts of silver with Ts increase the ionization of Ts [188]. While it is possible to produce positively and negatively charged ions during electrospray ionization (ESI), Lanina et al. [189] have showed that negative ionization is superior in terms of efficiency for the analysis of Ts. Negative ionization leads to the formation of deprotonated molecular ions without further fragmentation. To assist the formation of deprotonated ions, Bustamante-Rangel et al. [187] used a slightly alkaline mobile phase—6.0 mM ammonia in methanol/water (97:3, *v*/*v*). The limits of detection were 0.015 μg/mL, 0.0055 μg/mL, and 0.0062 μg/mL for α-T, γ-T, and δ-T, respectively. In the atmospheric pressure chemical ionization (APCI) technique, the sample is first converted into small droplets in a nebulizer by high-speed nitrogen flow. After desolvation in a heated vaporization chamber, the analytes are ionized via corona discharge and transferred to a mass analyzer [190]. Lanina et al. (2007) showed that APCI ionization in the negative mode was superior for the determination of tocochromanols in comparison with ESI, as it produced lower detection limits (2–6 times), a higher linearity range (7.5–25,000 ng/mL APCI (−), 15–3700 ng/mL ESI(−)) and proved to be more robust to the nature of solvents and structure of the analytes [189]. Viñas et al. [191] have been using the following APCI parameters for the determination of Ts: drying temperature 350 °C, APCI temperature 400 °C, drying gas flow 5 L min^−1^ and nebulizer gas pressure of 60 psi. In atmospheric pressure photoionization (APPI) techniques, the sample is first converted into gas phase by using a heated nebulizer, and after interacting with photons emitted by a discharge lamp and series of gas-phase reactions, the molecules are ionized. Méjean et al. [94] applied SFC-qTOF for the analysis of Ts and T3s using ESI, APPI and APCI ionization techniques and showed that the APPI source is more sensitive and robust in comparison with APCI.

The most conventional and simple mass analyzers are single quadruple (Q) low-resolution MS systems, which can be operated in full scan or selected ion monitoring (SIM) mode. In a paper by Lanina et al. [189], LC-APCI(-)-Q was used for the analysis of Ts in oil and milk samples, and quantification was based on external standards to construct the calibration curves in SIM mode. The LODs in this case were around 3 ng/mL, which is comparable to an FLD detector. The main disadvantage of single quadruple systems is limited information about structural formation. In studies where it is necessary to detect trace levels of tocochromanols and tocol derivatives in complex matrixes, triple quadruple (QqQ) systems can be used. Single reaction monitoring (SRM) mode has excellent sensitivity. An LC-APCI-QqQ system was successfully used by Nagy et al. [192] to identify tocochromanol-related compounds in human plasma. New studies are discovering more tocol derivatives (metabolites, oxidation products), and high-resolution MS (HRMS) techniques such as Time-Of-Flight (TOF) and Orbitrap mass analyzers are beneficial for the identification of these novel or minor compounds.

### 4.5. Chromatographical Determination of Tocomonoenols (T1s), Tocodienols (T2s) and Esterified Tocochromanols (ETs)

T1s, T2s, and tocochromanol esters have been detected in plant material in low and very low concentrations and therefore require special tools for their separation, detection and identification. For instance, α-T1 cannot be completely separated by RPLC even when using PFP columns with core–shell technology and small particles (2.6 μm), relatively long columns (150 mm) and high amount of water as mobile phase (20%) to improve the separation between the tocochromanols. α-T1 was eluted by RPLC between the β-T and γ-T, which makes the separation even more difficult (run time of analysis 40 min) [142]. GC-MS is very helpful in determination of T1s and T2s. α-T1 was completely separated from other tocochromanols by GC-MS, and the time required was nearly half shorter that of RPLC [142]. A much better solution for the separation of T1s and T2s, at least for α homologues, would be the application of NPLC in comparison to RPLC, where α-T1 and α-T2 are eluted between α-T and α-T3 (analysis run time 30 min for Ts, α-T1, α-T2, and T3s) [52]. The advantage of using NPLC for the separation of T1s (α-T1 and γ-T1) in pumpkin seed oil has been demonstrated as well [193]. NPLC has also been used for the separation of tocochromanols, including α-T1 in palm-derived T3 rich fraction [194]. However, the most impressive combination of both separation and analysis time (below 5 min for four Ts, α-T1, α-T2, and four T3s) has been obtained by UPC^2^ and a supercritical carbon dioxide (CO_2_):methanol (99.5:0.5 *v*/*v*) mixture as the mobile phase in isocratic flow [52]. The UPC^2^/SFC seems to be not only very promising but also a very powerful tool for the separation of T1s and T2s in comparison to other chromatographic techniques. The superficial and limited knowledge about the presence of T1s and T2s is associated with several challenges, including the lack of available standards and very low concentration. CCC is recommended for the enrichment of samples prior to the detection/identification of trace lipid molecules, which are challenging to identify without a sufficient concentration [16,73].

Studies about the determination of ETs are limited due to several reasons: little knowledge about ET presence in nature, inclusion of saponification in protocols (liberation of bound Ts), lack of methodologies/protocols for their extraction from the plant matrix and sufficiently sensitive, selective, specific, and rapid methods of their detection, and, finally, lack of standards. Although esterified tocochromanols were first described in the sixties [53], there are still few reports on this topic, while the literature about their determination without liberating from esterified forms is even more scarce, and all of them are published in the last 7 years with only one exception. Studies on the presence of ETs, tocopheryl esters and their relative abundance (%) in plant material were first conducted in 1994 using a GC-FID and GC-MS system [85]. Tocopheryl esters were identified by MS [85]. The same tools were still used over a decade later in 2016. To determine ETs by GC-FID, the standards of ETs were obtained by esterification with future purification by TLC and the identification by Nuclear Magnetic Resonance (NMR) spectroscopy. The separation of four ETs in heated sunflower oil by GC-FID was completed in 60 min [87]. In 2013, in residues and distillates of structured lipids, the ETs were identified using not only GC-MS in synchronous scan/SIM mode by employing two ionization modes, electron (EI) and chemical (PCI), but also MALDI-TOF-MS to confirm identification [89].

In 2017, ETs were separated from laboratory deodorized vegetable oil and detected not only by GC-FID and GC-MS, but also utilizing a RPLC-UV method for ET determination for the first time. The four ETs were separated by RPLC-UV using a C18 column with isocratic flow of methanol:2-propanol (65:35, v/v) in a 50 min chromatographic run at wavelength 284 nm [88]. The most detailed ET composition thus far was demonstrated in vegetables. For ET identification, standards were synthesized, and a GC-MS system was used in SIM mode for their determination in a 35 min chromatographic run [10,86]. More research is required to better understand the possibility of using techniques other than GC-MS, which presently seems to be the best option due to separation and identification issues.

## 5. Summary, Future Perspectives and Recommendations

To avoid misinterpretation of the obtained results each analytical technique should be implemented according to the purpose of the study.

Along with β homologues, other less common tocochromanols are generally not analyzed. Rare occurrence and low concentration in plant material is unhelpful for identification and detection, therefore, the suitable tool and protocol development for the determination of minor tocochromanols is required.

It is strongly recommended to use analytical tools that allow to separate the highest number of tocochromanols as possible, especially in investigation of non-reported plant material.

Alkaline hydrolysis (saponification) can be insufficient for the recovery of all tocochromanols in some plant materials, e.g., green coffee beans, while short thermal treatment, for instance, roasting, can cause the release/liberation of bound tocochromanols or formation of non-tocochromanol compounds.

For the analysis of plant material treated at high temperature, for instance, roasted beans and seeds, the application of different separation techniques and/or MS are advised to confirm ambiguous or inconclusive results, e.g., a significant increase in tocochromanols during treatment.

In reversed-phase separations, it is highly recommended to use columns capable of separating β and γ isomers (PFP/F5, C30, RP-amide, PYE and πNAP), especially for plant material which contains both isomers or has not been studied. Methanol and acetonitrile as the RPLC mobile phase do not have sufficient eluting strength to elute PC-8 from the column. Addition of *n*-hexane to the mobile phase is necessary to elute PC-8, however, it will result in the coelution of major tocochromanols.

Based on several reports, it should be advised to use NP and RP complementarily to provide a “true picture” of the tocochromanol content and profile, especially when results are ambiguous. The complementary use of both techniques could easily prevent serious misinterpretation of the results without using high-cost equipment, e.g., MS detection.

Supercritical fluid chromatography, the latest advancement in alternative tocochromanol analysis techniques, can be considered as a powerful tool for the separation of Ts, T3s and PC-8 as well as the less common T1s and T2s. It allows for the highly efficient, selective and environmentally friendlier separation of tocochromanols due to a combination of LC and GC features with the possibility of adjusting many separation parameters, such as temperature, back pressure, and type of column, allowing to reach NP or RP flow without changing the mobile phase, type of modifier(s), and the possibility of using a gradient both in relation to CO_2_ and between the mixture of modifiers.

However, the development of proper extraction and identification protocols is required for the determination of free, esterified, and non-extractable tocochromanols. Studies on the application of the alternative extraction and/or sample preparation techniques should be comparative and accompanied by conventional methods, e.g., saponification.

In the last decade, the topic of bound tocochromanols has returned, and more attention is being put on the occurrence of other minor tocochromanols such as T1s and T2s. The most common tool for the determination of these molecules is GC due to improved separation in comparison to LC, while SFC is becoming a good alternative to GC.

Although 100 years have passed since the discovery of the first tocochromanol (α-T, commonly referred to as vitamin E), there are still many unknowns. Modern tool and appropriate protocol application during tocochromanol determination will eliminate information gaps about these bioactive molecules.

## Figures and Tables

**Figure 1 molecules-27-06560-f001:**
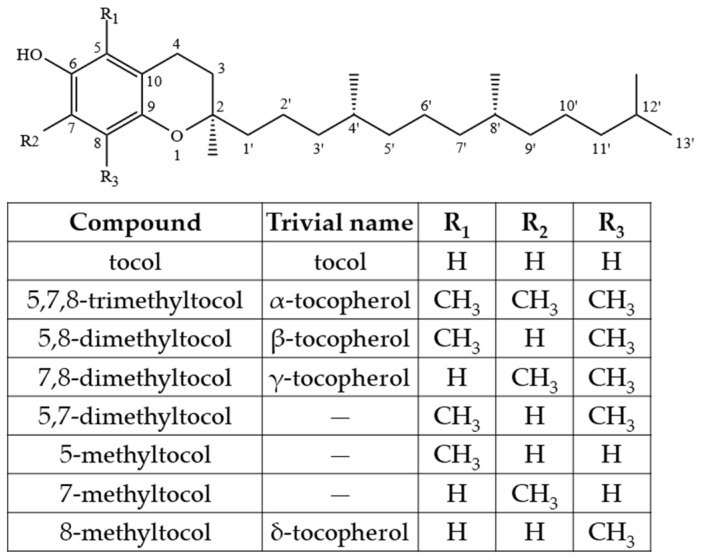
Chemical structures of tocol-related compounds.

**Figure 2 molecules-27-06560-f002:**
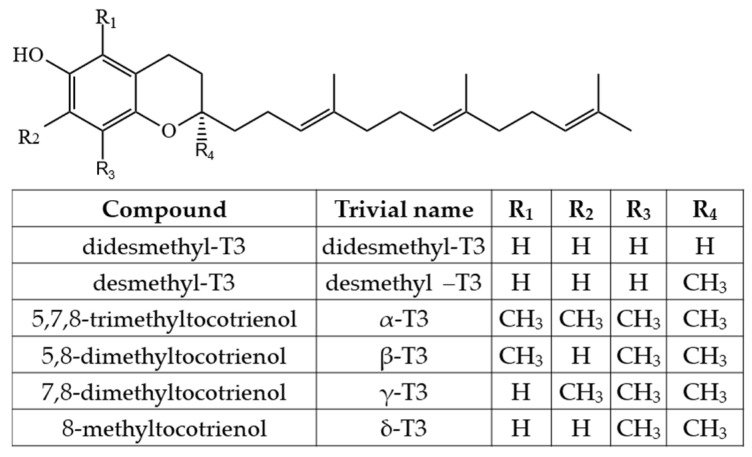
Chemical structures of tocotrienol-related compounds.

**Figure 3 molecules-27-06560-f003:**
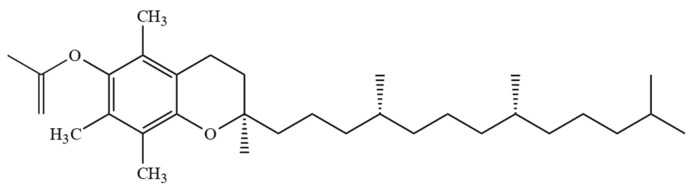
Chemical structure of α-tocopheryl acetate.

**Figure 4 molecules-27-06560-f004:**
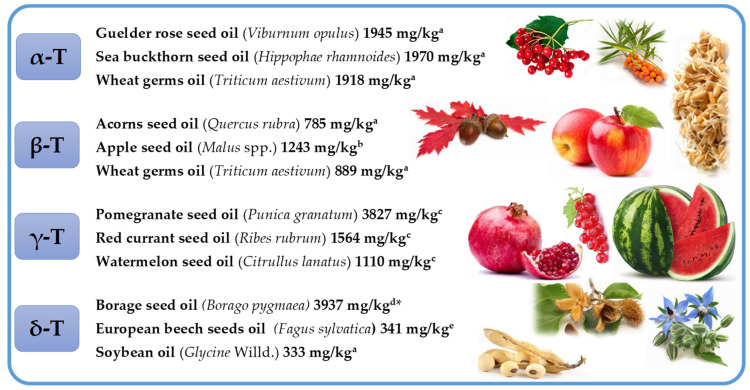
Selected richest sources of four tocopherol (T) homologues (α, β, γ, and δ). ***** Concentration of tocopherols was calculated as α-tocopheryl acetate equivalents. Superscripts (^a–e^) refers to the following literature positions: ^a^, [12]; ^b^, [42]; ^c^, [35]; ^d^, [46]; ^e^, [47].

**Figure 5 molecules-27-06560-f005:**
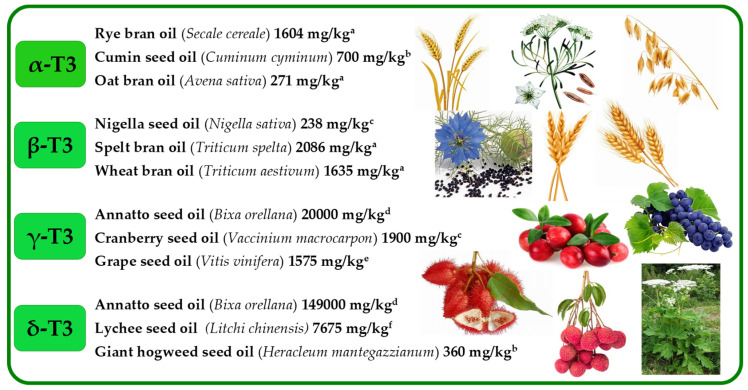
Selected richest sources of four tocotrienol (T3) homologues (α, β, γ, and δ). Superscripts (^a–f^) refers to the following literature positions: ^a^, [38]; ^b^, [50]; ^c^, [12]; ^d^, [54]; ^e^, [57]; ^f^, [63].

**Figure 6 molecules-27-06560-f006:**
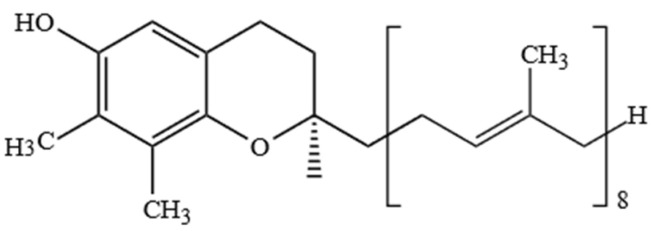
Chemical structure of plastochromanol-8.

**Figure 7 molecules-27-06560-f007:**
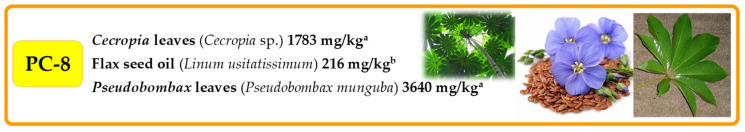
Selected richest sources of plastochromanol-8 (PC-8). Superscripts (^a–b^) refers to the following literature positions: ^a^, [65]; ^b^, [15].

**Figure 8 molecules-27-06560-f008:**
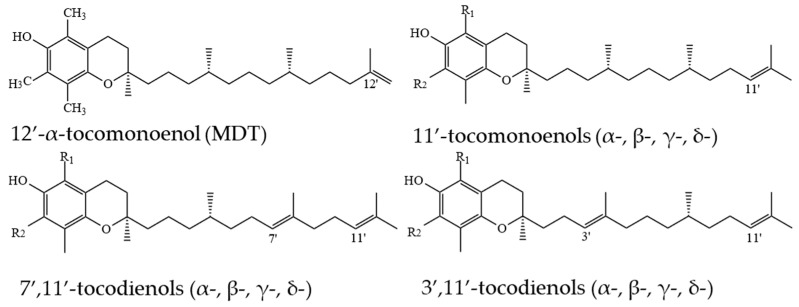
Chemical structure of tocomonoenols and tocodienols. For α-, β-, γ-, δ- methyl substituent number and position see Figure 1.

**Figure 9 molecules-27-06560-f009:**
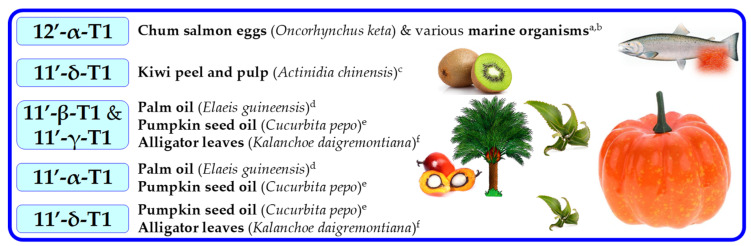
Sources of tocomonoenols (T1s). Superscripts (^a–f^) refers to the following literature positions: ^a^, [70]; ^b^, [71]; ^c^, [75]; ^d^, [73]; ^e^, [16]; ^f^, [74].

**Figure 10 molecules-27-06560-f010:**
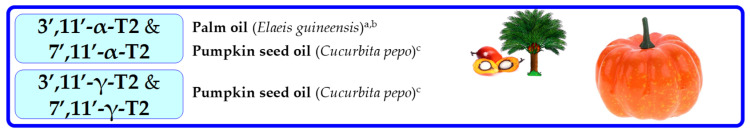
Sources of tocodienols (T2s). Superscripts (^a–c^) refers to the following literature positions: ^a^, [52]; ^b^, [73]; ^c^, [16].

**Figure 11 molecules-27-06560-f011:**
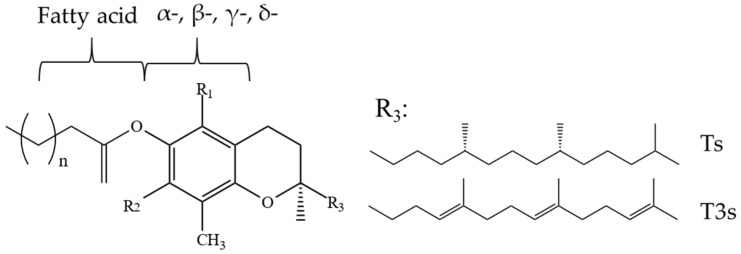
Chemical structure of tocochromanol fatty acid esters. For α-, β-, γ-, δ- methyl substituent number and position see Figure 1.

**Table 1 molecules-27-06560-t001:** Comparison of result interpretation of the tocopherol (T) and tocotrienol (T3) isomers (β and γ) determination in the same type of plant material by LC using different columns (C18, C30, PFP, and Diol) and elution order (NP and RP).

Common Name	Latin Name	Homologue Occurrence				Phase, Column	Ref.	Homologue Occurrence and β % of Σβ + γ						Phase, Column	Ref.
		β-T	γ-T	β-T3	γ-T3			β-T	γ-T	β-T %	β-T3	γ-T3	β-T3 %		
Black caraway	*Nigella sativa*	–	yes	-	yes	RP, C18	[37]	yes	yes	21%	yes	–	100%	NP, Diol	[51]
European cranberrybush	*Viburnum opulus*	–	yes	-	–	RP, C18	[56]	yes	yes	91–96%	–	–	–	RP, C30	[43]
Sea buckthorn	*Hippophaë rhamnoides*	–	yes	-	–	RP, C18	[56]	yes	yes	15–79%	yes	yes	9%	RP, PFP	[156]
Wheat	*Triticum* sp.	–	yes	-	–	RP, C18	[157]	yes	yes	92%	yes	-	100%	RP, PFP	[158]

- not detected.

## Data Availability

The data used to support the findings of this study are available in Appendix A and from the corresponding author upon request.

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
