# Peer review of "Free and Esterified Tocopherols, Tocotrienols and Other Extractable and Non-Extractable Tocochromanol-Related Molecules: Compendium of Knowledge, Future Perspectives and Recommendations for Chromatographic Techniques, Tools, and Approaches Used for Tocochromanol Determination"

_molecules, 2022, doi:10.3390/molecules27196560_

Round 1
Reviewer 1 Report
The review is well detailed and may be useful to scientists in the field of natural product chemistry associated with medicinal chemistry. In particular, I consider that it is an extensive but easy to understand review where the information is clear and precise.
Author Response
Thank you for the positive overview.
Reviewer 2 Report
The reviewed article is devoted to a review of current data on the advantages and disadvantages of various chromatographic methods, tools and approaches that are used to separate and detect tocochromanols (tocopherols, tocotrienols, plastochromanol-8, tocomonoenols and tocodienols) in plant material and foodstuffs. The paper discusses future problems and prospects, offers recommendations for the implementation of green methods for the determination of tocochromanols. The results of the work may be of interest both in terms of presenting the key aspects and protocols associated with the determination of tocochromanol, and for the correct identification and interpretation of the obtained outcomes. It is desirable to improve the quality of the Figure 2 and the article can be publication in its present form.
Author Response
Thank you for the positive overview. Figure 2 has been split up into several smaller figures, and the quality has been improved.
Reviewer 3 Report
Molecules
Review: "Free and esterified tocopherols, tocotrienols and other extractable and non-extractable tocochromanol-related molecules Compendium of knowledge on chromatographic techniques, tools, and approaches used for determination: future perspectives and recommendations".
- I suggest to add short note about the pharmacological activity and therapeutically use for tocopherols in addition for their antioxidant activity.
- Figure 2 must be improved
-LC-DAD: add the meaning in the abbreviation section
Author Response
Review: "Free and esterified tocopherols, tocotrienols and other extractable and non-extractable tocochromanol-related molecules Compendium of knowledge on chromatographic techniques, tools, and approaches used for determination: future perspectives and recommendations".
- I suggest to add short note about the pharmacological activity and therapeutically use for tocopherols in addition for their antioxidant activity.
Thank you for the comment. Since this review is already overladed and generally linked with the topic of identification, we would like to skip the addition of information about pharmacological activity and therapeutically use for tocopherols. About the tocopherols is known much more that other tocochromanol forms e.g. tocotrienols and esterified tocochromanols. Therefore, we have already included in the introduction part: “the although several reports indicate superior biological properties of tocotrienols to tocopherols, such as antioxidant and anti-inflammatory activities as well as their preventive effect against cancer, diabetes, cardiovascular and neurodegenerative diseases [6-8]. In the case of α-T, there is a clear lack of cancer-preventive activity. As for δ-, γ-T, and tocotrienols, the effect on cancer risk is at least unclear and requires more studies [9]. Another topic worth highlighting is ester-bound tocochromanols, which are non-fissile in the human gastrointestinal tract and their potential biological activity is unknown [10].” We believe that it is enough for the current review topic.
- Figure 2 must be improved
Thank you for the comment. The Figure 2 has been split up in several smaller figures, and the quality has been improved.
-LC-DAD: add the meaning in the abbreviation section.
Thank you for the comment. LC-DAD abbreviation, as well as other abbreviations used in the manuscript, were added.
Additionally we did:
- improved literature about the esterified bounded tocochromanols (one reference has been added);
- NPLC, RPLC and SFC tables with chromatographically conditions were moved to supplementary materials to reduce the number of pages and to make the manuscript and tables more readable for the potential reader;
- one reference was removed;
- some English spelling/grammar were revised.
Reviewer 4 Report
A more brief title should be insered.
The novelty character of this review respect to the other present in literature
should be better marked.
The distinction between extractable and non extractable molecules should be described in the text and related references added.
The distribution of compounds in food groups shoul be better described
Innovative and future directions should be better discussed.
Author Response
A more brief title should be inserted.
Thank you for the comment. Well, we were thinking and trying several titles, and we think that the present one in "brief" gives a full description of what can be found in a present review paper. Yes, it is a bit long, but it is very informative for the potential reader, and we think that this is the most important, however we still did a slight changes.
The novelty character of this review respect to the other present in literature should be better marked.
Thank you for the comment. We added in the abstract “Procedures for obtaining different tocochromanol analytical standards are also described.” In abstract is also included already “The presence of free, esterified, and non-extractable tocochromanols in plant material …” and “This review discusses the advantages and disadvantages of different chromatographic techniques, tools, and approaches used for the separation and detection of different tocochromanols in plant material and foodstuffs” and “Finally, future challenges, perspectives, recommendations, and green techniques implementation for tocochromanol determination are discussed and proposed”. The issues of free, esterified, and non-extractable tocochromanols, obtaining of minor tocochromanol-related analytical standards, and green techniques implementation for tocochromanol determination were not described in any review papers. Additionally in the Introduction part it has been added: “This paper discusses the different tools, methods, and techniques for the determination of major (Ts, T3s, and PC-8), minor (tocodienols (T2s), tocomonoenols (T1s)), and less described esterified tocochromanols. The issue of extractable and non-extractable tocochromanols, methods for obtaining analytical standards of tocochromanols, and environmentally friendlier approaches was also discussed”.
The distinction between extractable and non extractable molecules should be described in the text and related references added.
Thank you for the comment. The thing is that, as we mentioned in the manuscript: "The topic of extractable and non-extractable tocochromanols has not been investigated". Lack even used such terms, as we mentioned in the manuscript "The terms ‘Non-extractable tocochromanols’, ‘non-extractable tocopherols’, ‘non-extractable tocotrienols’ are not present in scientific literature". In the paragraph "Non-extractable tocochromanols" we described all that could be linked with the term "non-extractable tocochromanols".
The distribution of compounds in food groups should be better described
Thank you for the comment. Since this review is already overloaded and generally linked with the topic of identification not sources of tocochromanols, we do not think it is necessary to focuses more on this topic. We present in five figures, and nine tables in supplementary material the food/plant sources of specific tocochromanols.
Innovative and future directions should be better discussed.
Thank you for the comment. Well, I think all is clear and provided in the part "Summary, future perspectives, and recommendations". The future direction can be using SFC as well as continuing GC-MS, which is also already highlighted in that part. Additionally, we added a few reports about esterified tocochromanols to which we did not have access previously. We believe that we included all reports with less spread in nature and less described tocochromanols which can be a future direction. It is included: "there are still many unknowns. Applying modern tools and appropriate protocols during tocochromanol determination supports the filling of information gaps about these bioactive molecules". Finally, we believe that the aspect of future directions is enough well described.
Additionally we did:
- improved literature about the esterified bounded tocochromanols (one reference has been added);
- NPLC, RPLC and SFC tables with chromatographically conditions were moved to supplementary materials to reduce the number of pages and to make the manuscript and tables more readable for the potential reader;
- one reference was removed;
- some English spelling/grammar were revised.
Reviewer 5 Report
Very nice work.
Just a couple technical suggestions.
All are listed below with an appropriate Line number(s) from text in order to facilitate tracking:
Line 20: Move "of" in front of "great" here.
Line 44: Suggest to replace "Since this" with "Since then". It seems to me much more appropriate in this context.
Line 48: Suggest to move the first "vitamin E" to the Line 47 in front of "ataxia", and to delete "with and the first "deficiency" in the Line 48.
Line 146: Maybe "lacked" will be better choice here instead of "not found"?
Line 163: Missing "max." in the Latin name for soybean. Correct.
Line 198: Be consistent - either both "crowberries" and "cranberries" in plural or both in singular. Correct.
Line 199: typo - surplus "to the". Delete one.
Line 205: Missing "of" after "leaves" here.
Line 384: "It was demonstrated..." Please correct.
Line 759: Put term "bis" in Italic here
Line 909: typo - why bolded part of text in this Line? Please check.
Kind regards.
Author Response
Very nice work.
Thank you for the positive overview.
Just a couple technical suggestions.
All are listed below with an appropriate Line number(s) from text in order to facilitate tracking:
Line 20: Move "of" in front of "great" here.
Thank you. The correction has been done according to the recommendation.
Line 44: Suggest to replace "Since this" with "Since then". It seems to me much more appropriate in this context.
Thank you. The correction has been done according to the recommendation.
Line 48: Suggest to move the first "vitamin E" to the Line 47 in front of "ataxia", and to delete "with and the first "deficiency" in the Line 48.
Thank you. The correction has been done according to the recommendation.
Line 146: Maybe "lacked" will be better choice here instead of "not found"?
Thank you. We changed it to “does not occur”.
Line 163: Missing "max." in the Latin name for soybean. Correct.
Thank you. Since in the cited reference was "Glycine Wild." we would like to keep it as it is.
Line 198: Be consistent - either both "crowberries" and "cranberries" in plural or both in singular. Correct.
Thank you. The correction has been done according to the recommendation.
Line 199: typo - surplus "to the". Delete one.
Thank you. The correction has been done according to the recommendation.
Line 205: Missing "of" after "leaves" here.
Thank you. The correction has been done according to the recommendation.
Line 384: "It was demonstrated..." Please correct.
Thank you. The correction has been done according to the recommendation.
Line 759: Put term "bis" in Italic here
Thank you. The correction has been done according to the recommendation.
Line 909: typo - why bolded part of text in this Line? Please check.
Thank you. The correction has been done according to the recommendation.
Additionally we did:
- improved literature about the esterified bounded tocochromanols (one reference has been added);
- NPLC, RPLC and SFC tables with chromatographically conditions were moved to supplementary materials to reduce the number of pages and to make the manuscript and tables more readable for the potential reader;
- one reference was removed;
- some English spelling/grammar were revised.